# SELF-DISCRIMINATIVE OPTIMIZATION FOR VIDEO DIFFUSION MODELS

## ABSTRACT

Recent preference alignment strategies have gained traction in large language models (LLMs) and are now being extended to broader generative domains. Approaches such as Direct Preference Optimization have been adapted to diffusion models by leveraging human-labeled preferences or auxiliary score models to distinguish "winner" from "loser". However, these methods face two key challenges: (1) the optimization process often overfits to the score model, resulting in suboptimal generation quality; and (2) the results generated from the same text prompt exhibit significant divergence, resulting in limited effective gradients and reduced training efficiency. These limitations are further exacerbated in video generation, where evaluation is more complex and inference is slower. In this work, we introduce Self-Discriminative Optimization that using only a handful of real samples, unlocks markedly higher-quality generation. First, we introduce self-degradation that applies frequency-domain reweighting to the latent representations from real samples, yielding degraded samples that more closely match the model's original output distribution. This leads to controlled distortions such as low-quality, temporal inconsistency and object deformation. We then use these real/degraded pairs as positive and negative examples to fine-tune the pretrained model discriminatively with automatically assigned, reliable labels. By exploiting the richer gradients from these controllable degradation pairs, our experiments demonstrate substantial gains in structural quality and semantic alignment using only a handful of high-quality samples and minimal fine-tuning.

## 1 INTRODUCTION

Text-to-video (T2V) generation models Blattmann et al. (2023); Chen et al. (2023; 2024); Yang et al. (2024); Zheng et al. (2024); Kong et al. (2024); Wang et al. (2025) have gained significant attention due to their potential in various creative and industrial applications, such as automated filmmaking, content creation Kong et al. (2024); Wang et al. (2025). By translating natural language descriptions into coherent and visually rich videos, these models promise to revolutionize how media is produced and consumed. However, despite rapid advances in generative modeling, producing high-quality videos remains a challenging task. Compared to image generation, video generation demands not only spatial fidelity but also temporal consistency and motion realism, making the task inherently more complex and prone to failure in structure, coherence, or semantic alignment Huang et al. (2024a).

Recently, researchers have explored preference-based optimization for post-training large language and generative models. For instance, Direct Preference Optimization (DPO) Rafailov et al. (2023) has shown promise in aligning language models using human-labeled preferences or auxiliary scoring models. However, applying DPO to diffusion models introduces new challenges: (1) the optimization tends to overfit the learned score model, leading to poor generalization and suboptimal results; and (2) the long inference time of diffusion models, which requires multiple samples per prompt during training, severely limits scalability. These challenges become even more critical in video generation, where evaluation involves multiple interdependent dimensions such as spatial quality, temporal consistency, motion realism, and semantic alignment. This makes it significantly harder to define and optimize a *balanced preference signal*. The inherently slower inference speed of video diffusion models further compounds the problem. These video-specific obstacles substantially increase the complexity of model optimization and render direct extensions of alignment strategies from image or language models inadequate. More recently, Direct Discriminative Optimization (DDO) Zheng et al.

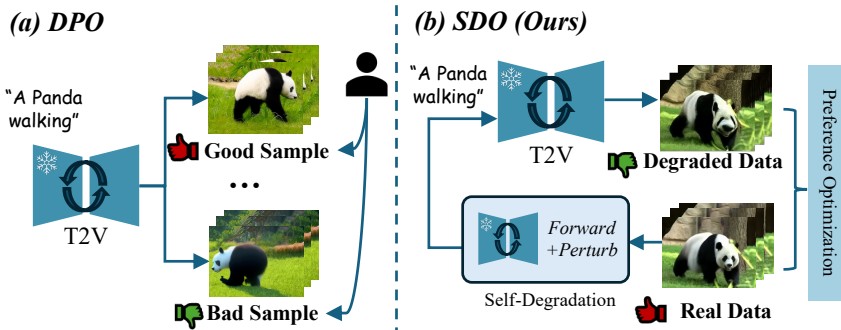

Figure 1: Compared to DPO Rafailov et al. (2023); Wallace et al. (2024), which relies on human-annotated good and bad samples, our SDO leverages self-degradation to automatically construct real/degraded data pairs, using them as positive and negative examples for reliable supervision. This enables efficient and scalable preference alignment in video generation.

(2025) has demonstrated that real data can also be leveraged to optimize generative models at a lower training cost than supervised fine-tuning, effectively guiding the model toward higher-quality regions of the data manifold. However, due to the use of unpaired data during training, DDO suffers from instability. The training process is prone to gradient vanishing, as the model can too easily distinguish real from fake pairs, leading to a lack of meaningful learning signals.

In this work, we propose a novel method Self-Discriminative Optimization that leverages ground-truth real data while stabilizing model optimization. Specifically, we first introduce a technique called self-degradation to construct real-fake data pairs. This approach eliminates the need for human annotations or external ranking models. Our key insight is to leverage structured degradations in the latent space as a proxy for low-quality generations. We perturb latent representations through frequency-domain reweighting. We empirically find that this process introduces controlled distortions, such as reduced visual quality, temporal inconsistency, and object deformation, while keeping the samples within the data distribution learned by the model. The resulting real/degraded pairs provide a reliable training signal to align with the real data manifold, with automatically assigned labels that reflect quality differences without human supervision. We then optimize the model to distinguish between real and fake pairs. Notably, these real and fake pairs are semantically correlated but differ in their levels of distortion. This design guides the model toward the real data distribution from the current fake distribution, while maintaining stable training dynamics. Finally, our approach enables efficient and scalable preference alignment in diffusion-based video generation models. Through experiments on CogVideoX-2B, CogVideoX-5B Yang et al. (2024), and Wan2.1-1.3B Wang et al. (2025), we demonstrate that our method significantly reduces motion artifacts, structural distortions, temporal inconsistencies, and other common failure cases in video generation, using only a small amount of high-quality training data and minimal fine-tuning. Our main contributions are summarized as follows:

- *Self-degradation*. We propose a controlled degradation synthesis method that leverages the reference diffusion model by introducing frequency perturbations to the latent space derived from real videos. This process simulates degraded data that remains within a data distribution similar to what the model has already learned.

- *Self-discriminative optimization*. We introduce a label-free optimization framework that directly fine-tunes the model to distinguish between real and degraded counterparts, providing a stable and effective optimization guidance without relying on explicit preference labels or reward models.

- *State-of-the-art optimization gains with minimal cost*. Our proposed optimization approach consistently improves performance across all evaluated base models compared to existing post-training methods, while requiring only a small amount of training data and a limited number of tuning steps, without relying on human annotations or additional score models.

## 2 RELATED WORK

### 2.1 TEXT-TO-VIDEO DIFFUSION MODELS

With the advancement of diffusion models Ho et al. (2020), text-to-video generation Wang et al. (2023); He et al. (2022); Zhang et al. (2023); Blattmann et al. (2023); Chen et al. (2023; 2024) has garnered increasing attention for its ability to produce visually compelling content that accurately reflects textual input, ultimately aiming to fulfill user intent. The underlying architectures have evolved from traditional convolutional U-Net designs Blattmann et al. (2023); Chen et al. (2023; 2024) to more recent transformer-based diffusion backbones Yang et al. (2024); Zheng et al. (2024). To ensure temporal consistency across frames, researchers have adopted mechanisms such as temporal attention, transformer layers, and 3D convolutional blocks. However, due to the inherent complexity of video data, adapting diffusion pipelines for high-quality video generation presents significant challenges. It still may lead to issues such as temporal inconsistency, inaccurate motion, and object deformation. This challenge is compounded by the necessity of implementing a series of post-processing or post-training methods aimed at enhancing video quality of base video generation models. For post-processing, VEnhancer He et al. (2024) enhances generated videos by introducing additional forward processes and retraining enhancement modules using paired data. In contrast, FreeInit Wu et al. (2024) adopts a training-free approach, applying the diffusion model in multiple forward passes to improve video quality. They leverage multiple forward passes over time to improve generation quality. More recently, inspired by the success of alignment optimization techniques in large language models, several works have incorporated human preferences or external scoring models to guide generation post-training Yuan et al. (2024); Liu et al. (2024).

### 2.2 DIRECT PREFERENCE LEARNING IN DIFFUSION MODELS

Reinforcement Learning from Human Feedback (RLHF) Christiano et al. (2017); Ziegler et al. (2019) has become a core method for aligning models with human preferences, widely adopted in both large language models Ouyang et al. (2022) and diffusion models Xu et al. (2023); Fan et al. (2023). Among RLHF methods, Direct Preference Optimization (DPO) Rafailov et al. (2023) stands out for its simplicity and effectiveness, enabling reward-free alignment that fits well with diffusion models. DiffusionDPO Wallace et al. (2024) adapts DPO to the diffusion framework via a differentiable objective, while VideoDPO Liu et al. (2024) extends it to video generation using a video scoring model for preference ranking. Other approaches Prabhudesai et al. (2024); Li et al. (2024) employ reward models directly, such as T2V-Turbo Li et al. (2024), which optimizes single-step rewards during distillation. However, video benchmarks involve complex and entangled dimensions, making optimization prone to overfitting on score models or annotations. To mitigate this, Zheng et al. Zheng et al. (2025) proposed Direct Discriminative Optimization (DDO), showing promising results, though its unpaired training may lead to instability and gradient vanishing in video settings with limited data. In this work, we introduce a *label-free optimization* framework based on a degrade-then-optimize paradigm. By learning from *self-degraded* examples, our method aligns model outputs without relying on explicit labels or handcrafted metrics, using only a small amount of training data.

## 3 PRELIMINARY

**Video Diffusion Models.** Diffusion models synthesize visual content by progressively removing random noise through a series of denoising steps. Starting from a noise vector $\mathbf{z}_T$, the model estimates cleaner versions at each step to finally generate the desired output $\mathbf{z}_0$. The forward diffusion process corrupts the data over time via:

$$\mathbf{z}_t \sim \mathcal{N}(\sqrt{\gamma_t}\,\mathbf{z}_{t-1}, \sigma_t^2\mathbf{I}),$$

where $\gamma_t = 1 - \sigma_t^2$ controls the signal preservation, and $\sigma_t^2$ determines the variance of the noise introduced at timestep $t$.

The denoising network $\hat{\epsilon}_\theta$ learns to reverse this process by predicting the noise component added at each step. The training objective minimizes the prediction error against the true noise:

$$\mathcal{L}_{\text{diff}}(\theta) = \mathbb{E}_{t,\mathbf{z}_0,\epsilon}\left[\left\|\epsilon - \hat{\epsilon}_\theta\left(\sqrt{\bar{\gamma}_t}\,\mathbf{z}_0 + \sqrt{1 - \bar{\gamma}_t}\,\epsilon, t\right)\right\|^2\right],$$

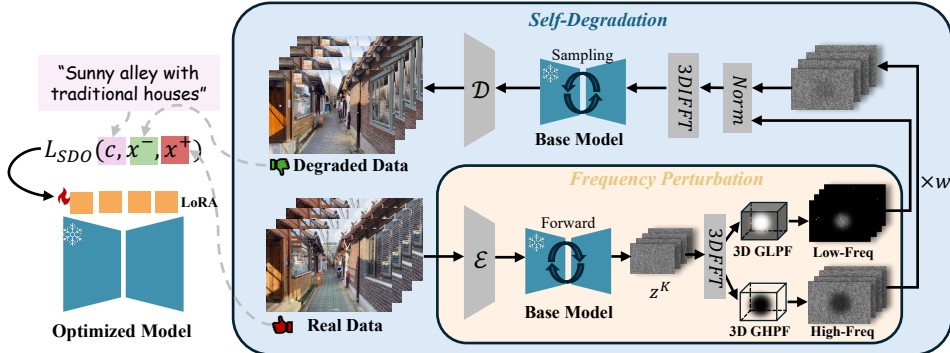

Figure 2: **Overview of our self-discriminative optimization (SDO).** Real data is sampled from a small, high-quality video dataset and forwarded to a noisy latent of timestep $K$, producing $\mathbf{z}^K$. A 3D FFT is applied to decompose $\mathbf{z}^K$ into low- and high-frequency components, where the high-frequency part is reweighted by a factor $w$ to introduce perturbation. The perturbed and normalized latent is then used to perform diffusion sampling, resulting in degraded data. The real data, degraded data, and associated text prompt are jointly used to train LoRA adapters integrated into the base model.

where $\bar{\gamma}_t = \prod_{s=1}^{t} \gamma_s$ is the cumulative product of signal retention factors over time.

**Diffusion DPO.** Direct Preference Optimization (DPO) Rafailov et al. (2023) is a method designed to align generative models with human preferences by learning from pairs of outputs labeled as positive (preferred) and negative (less preferred). The model is optimized to favor generating samples similar to the preferred ones.

In the diffusion model context, DiffusionDPO Wallace et al. (2024) applies this concept to guide image synthesis. The learning objective compares the model's response to both types of samples:

$$\mathcal{L}_{\mathrm{DPO}} = -\mathbb{E}_{(\mathbf{c}, \mathbf{z}^+, \mathbf{z}^-)} \left[ \log \sigma \left( L_\theta(\mathbf{z}^+, \mathbf{c}) - L_\theta(\mathbf{z}^-, \mathbf{c}) \right) \right],$$

where $\mathbf{z}^+$ and $\mathbf{z}^-$ represent the preferred and non-preferred samples, respectively, and $L_\theta(\mathbf{z}, \mathbf{c}) := \beta \log \frac{p_\theta(\mathbf{z}|\mathbf{c})}{p_{\theta_{\mathrm{ref}}}(\mathbf{z}|\mathbf{c})}$ denotes the model's conditional loss given input $\mathbf{z}$ and context $\mathbf{c}$, in which an approximation Wallace et al. (2024) is adopted for $\log \frac{p_\theta(\mathbf{z}|\mathbf{c})}{p_{\theta_{\mathrm{ref}}}(\mathbf{z}|\mathbf{c})}$ due to the computational cost as follows

$$\log \frac{p_\theta(\mathbf{z}|\mathbf{c})}{p_{\theta_{\mathrm{ref}}}(\mathbf{z}|\mathbf{c})} \approx \mathbb{E}_{t,\epsilon} \left[ -m(t) \left( \|\epsilon_\theta(\mathbf{z}_t, \mathbf{c}, t) - \epsilon\|_2^2 - \|\epsilon_{\theta_{\mathrm{ref}}}(\mathbf{z}_t, \mathbf{c}, t) - \epsilon\|_2^2 \right) \right]. \tag{1}$$

This contrastive formulation encourages the model to improve generation quality in accordance with user-aligned preferences.

**DDO.** Direct Discriminative Optimization (DDO) Zheng et al. (2025) focuses on aligning the model's output distribution with the ground-truth data distribution, offering an alternative to the traditional maximum likelihood estimation (MLE). By bridging likelihood-based generative models and GAN-like objectives, DDO enables direct optimization using the data manifold without requiring explicit supervision or paired comparisons. It relies solely on the original training data, which are unpaired with model-generated samples, making it both efficient and scalable.

$$\mathcal{L}_{\mathrm{DDO}} = -\mathbb{E}_{p_{data}(\mathbf{z}, \mathbf{c})} \left[ \log \sigma \left( L_\theta(\mathbf{z}, \mathbf{c}) \right) \right] - \alpha \mathbb{E}_{p_{\theta_{ref}}(\mathbf{z}, \mathbf{c})} \left[ \log \left( 1 - \sigma \left( L_\theta(\mathbf{z}, \mathbf{c}) \right) \right) \right],$$

where $\alpha$ is a balancing weight. This formulation guides the model toward the most reliable and high-fidelity regions of the data manifold.

## 4 METHODOLOGY

Inspired by DDO, we propose self-discriminative optimization, a novel degrade-then-optimize strategy that generates controllable degradations to deliver a stronger, more consistent training signal for perceptual enhancement.

## 4.1 SELF-DEGRADATION

We observe that artifacts in generated videos are primarily concentrated in the high-frequency components of the power spectrum. To synthesize realistic degraded videos, rather than suppressing these noisy components with low-pass filters—as commonly done to improve generation quality—we adopt the opposite strategy: amplifying high-frequency components during sampling to produce degraded outputs that remain consistent with the base model's generative distribution.

Specifically, we apply a frequency-domain-based perturbation to the intermediate latent representation in the diffusion process as shown in Figure 2. Given a clean latent representation $\mathbf{z}_0$, which is obtained by encoding a sample from the real dataset using the video encoder, we first obtain its noisy counterpart at step $K$ via the forward diffusion process:

$$\mathbf{z}_K = \sqrt{\bar{\alpha}_K}\mathbf{z}_0 + \sqrt{1 - \bar{\alpha}_K}\,\epsilon \tag{2}$$

where $\bar{\alpha}_K$ is the noise schedule at step $K$, and $\epsilon$ denotes the sampled Gaussian noise.

To apply spatial-temporal degradation, we transform $\mathbf{z}_K$ into the frequency domain using a 3D Fast Fourier Transform (FFT), and apply a modulated low-pass filter:

$$\tilde{\mathcal{F}} = \mathcal{FFT}_{3D}(\mathbf{z}_K) \odot (\mathcal{H} + w(1 - \mathcal{H})) \tag{3}$$

$$\mathbf{z}'_K = \Re\left[\mathcal{IFFT}_{3D}\left(\tilde{\mathcal{F}} \cdot \sqrt{\frac{\|\mathcal{F}_{\mathbf{z}_K}\|_2^2}{\|\tilde{\mathcal{F}}\|_2^2 + \varepsilon}}\right)\right] \tag{4}$$

Here, $\Re[\cdot]$ denotes taking the real part of the complex-valued output, $\mathcal{FFT}_{3D}$ and $\mathcal{IFFT}_{3D}$ denote the 3D Fast Fourier Transform and its inverse, operating across both spatial and temporal dimensions. $\mathcal{H}$ is a spatial-temporal low-pass filter that suppresses high-frequency details, while $w \in [1, 1.5]$ controls the degradation strength. The normalization term ensures energy consistency after filtering, and $\varepsilon = 1 \times 10^{-6}$ is a small constant to avoid division by zero. This self-degradation introduces realistic visual artifacts as shown in Figure 3 that commonly appear in synthesized videos, including the following:

- **Visual quality degradation:** Synthesized frames often exhibit noticeable artifacts such as over-smoothing, blurry regions, or unnatural textures. These issues reduce the overall visual fidelity of the video and make it appear less realistic or aesthetically pleasing to human observers.

- **Temporal unsmoothness:** Generated videos frequently lack temporal coherence, resulting in frame-to-frame inconsistencies such as flickering, abrupt transitions in motion, or unstable object appearances. For instance, the tail of a zebra may unnaturally intersect or pass through the body and then abruptly disappear, highlighting a failure to maintain consistent motion across frames.

- **Object shape distortion:** Objects within synthesized videos may undergo unrealistic geometric deformations, especially when subjected to motion or viewpoint variations. These distortions compromise both semantic and structural accuracy. For example, the shape of a zebra may appear stretched, compressed, or otherwise unnatural, leading to a visually implausible or confusing depiction.

## 4.2 TIMESTEP-AWARE SELF-DISCRIMINATIVE OPTIMIZATION

Unlike text-to-image generation, T2V generation often suffers from relatively lower quality, making it easier for the model to distinguish real from fake videos and thus more prone to gradient vanishing. To improve training stability and better align the optimization dynamics with the evolving state of the diffusion model, a timestep-aware adaptive scaling factor $\beta(t)$ is introduced. Unlike fixed-scale preference optimization, the adaptive $\beta$ increases with the diffusion timestep $t$, reflecting the intuition that later steps in the generation process require finer-grained adjustments, while earlier steps tolerate more aggressive optimization. This design allows the model to prioritize coarse preference corrections at early stages and refine preferences more delicately in later stages. The updated SDO loss under this adaptive weighting scheme is formulated as:

$$\mathcal{L}_{\text{SDO}}(\theta) = -\log\sigma(\beta(t) \cdot (L_\theta(\mathbf{z}^-) - L_\theta(\mathbf{z}^+))),$$

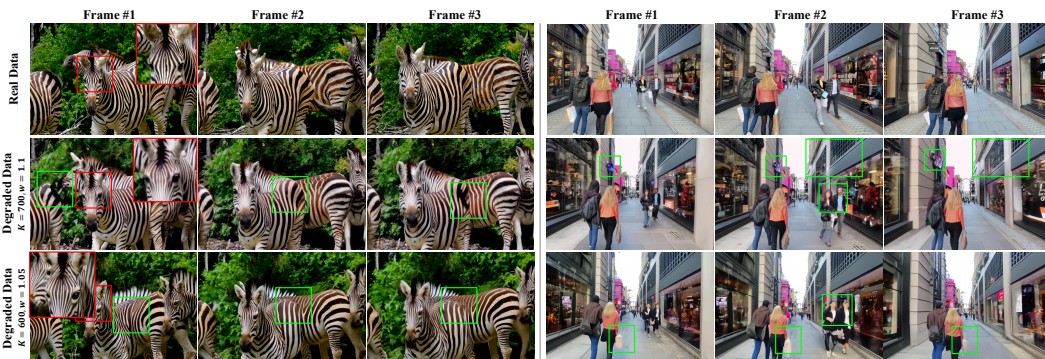

Figure 3: **Visualization examples of the degraded data and corresponding real data.** The proposed self-degradation is applied directly through the base model itself to intentionally degrade the data along several key dimensions: *visual quality* (e.g., overall reduction in visual quality), *temporal consistency* (e.g., the billboard changes from green to blue in degraded data 1 of the right example), *motion accuracy* (e.g., the swinging zebra tail passes through the zebra's body in degraded data 1 of the left example, or the two zebras merge and then split apart in degraded data 2 of the left example), *object shape distortion* (e.g., the zebra's body exhibits unnatural deformation in degraded data 1 of the left example), and *semantic decomposition* (e.g., the woman's bag is not held in her hand but instead floats in the air in degraded data 2 of the right example).

where $\sigma(\cdot)$ is the sigmoid function, and $L_\theta(x)$ represents the model's predicted loss for a given sample $x$ as defined in Sec. 3. Here, $\mathbf{z}^+$ and $\mathbf{z}^-$ refer to the preferred (winner) and dispreferred (loser) samples obtained as follows

$$\mathbf{z}^+ = \mathcal{E}(\mathbf{x}), \quad \mathbf{z}^- = \text{Rev}_{\theta_{ref}}(\mathbf{z}'_K), \tag{5}$$

where $\text{Rev}_{\theta_{ref}}$ is the sampling process of the pretrained video diffusion model, and $\mathbf{z}'_K$ is the perturbed latent feature elaborated in 4.1. The use of $\beta(t)$ ensures that the preference signal is dynamically scaled according to the generative timestep, enabling more stable and effective optimization throughout training. This adaptive modulation helps mitigate issues such as gradient explosion or collapse, and provides a principled way to integrate the temporal structure inherent to diffusion-based generation.

## 4.3 ANALYSIS AND DISCUSSION

Compared to existing diffusion-based DPO and DDO frameworks for high-fidelity image and video generation Wallace et al. (2024); Liu et al. (2024); Zheng et al. (2025), our proposed method adopts a degrade-then-optimize paradigm that requires only a small amount of training data, without relying on human annotations or additional score models. We also include comparisons with DPO and DDO in Section 5.2. This approach offers the following key advantages:

- **Self-degrade and self-optimize to close the domain gap.** The degraded data is generated directly by the reference diffusion model itself, effectively shifting overly refined or highly aligned outputs back toward its natural generative distribution. This enables optimization to proceed within a distribution the model is more familiar with, guiding it in a direction that aligns better with its intrinsic generation capabilities.

- **Reduce sampling efforts.** By leveraging real data, our method eliminates the need for multi-time sampling as required in DPO. A single forward pass of the self-degradation is sufficient to simulate the degraded data.

- **Avoid subjectivity and human bias.** Our method does not require human-annotated scores or handcrafted ranking models, which are often prone to subjective judgments and inconsistencies. Instead, it adopts a fully label-free approach, ensuring the optimization is guided purely by the model's own degradation-recovery process.

- **Improved training stability.** Unlike DDO, which uses unpaired training data and may suffer from instability or gradient vanishing, especially in video settings with limited samples,

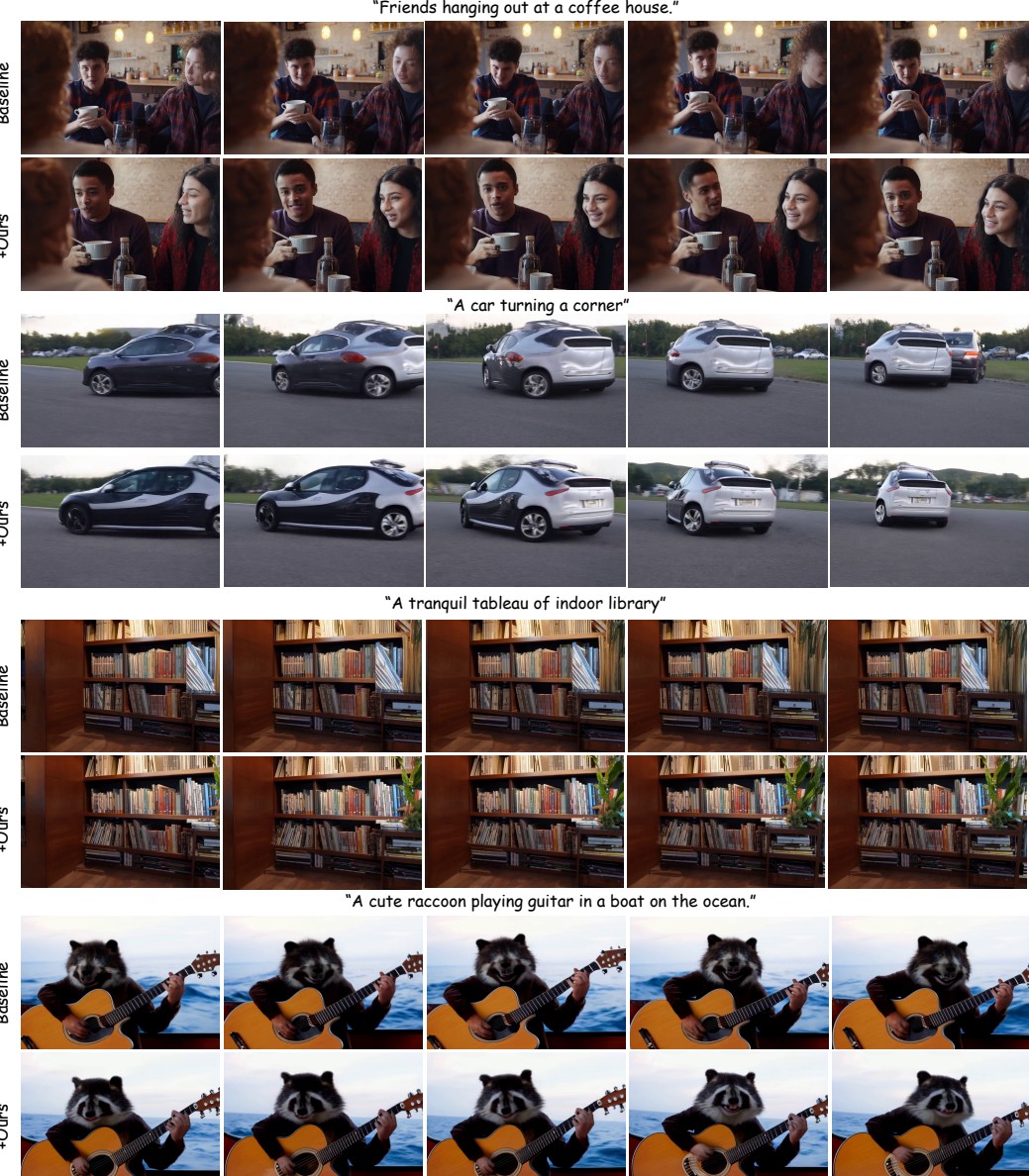

Figure 4: **Visual comparison.** Compared to the base model (CogVideoX-5B), our optimized version produces more consistent motion, fewer structural errors, resolves car composition issues, and generates richer details with more saturated colors.

our method offers more stable tuning. The complexity of video content makes it easier to distinguish real from fake, which amplifies this issue in DDO. In contrast, our approach avoids this by learning from paired degraded and real data.

## 5 EXPERIMENTAL RESULTS

### 5.1 EXPERIMENT SETUP

**Implementation Details.** We compare our pipeline with several state-of-the-art open-source text-to-video generation models, such as CogVideoX-2B Yang et al. (2024), CogVideoX-5B Yang et al. (2024), and Wan2.1-1.3B Wang et al. (2025), which are selected to assess performance across models of different scales. Our optimization approach can be applied to any text-to-video (T2V) base

| Model | Total | Motion smooth. | Dynamic degree | Aesthetic quality | Object class | Multiple objects | Human action | Spatial relation. | Scene | Appear. style | Subject consist. | Back. consist. |
|---|---|---|---|---|---|---|---|---|---|---|---|---|
| CogV-2B | 70.48 | 92.93 | 38.89 | 60.64 | 77.92 | 46.41 | 85.31 | 52.50 | 51.53 | 80.25 | 95.16 | 93.69 |
| +LoRA Tuning | 64.23 | 88.87 | 38.89 | 52.80 | 69.77 | 30.86 | 82.94 | 42.17 | 36.24 | 78.60 | 94.64 | 90.80 |
| +DDO | 69.02 | 89.05 | 37.86 | 55.32 | 74.93 | 50.06 | 83.27 | 55.18 | 45.76 | 78.61 | 94.04 | 95.19 |
| +VideoDPO[1] | 70.01 | 88.64 | 38.89 | 58.64 | 77.22 | 54.04 | 81.00 | 54.90 | 45.69 | 79.73 | 94.67 | 96.64 |
| +Ours | **71.09** | 93.62 | 38.89 | 57.95 | 79.35 | 52.13 | 85.44 | 60.31 | 47.28 | 79.46 | 95.12 | 92.42 |
| CogV-5B | 72.09 | 90.29 | 42.36 | 61.77 | 82.91 | 57.54 | 87.87 | 53.66 | 48.52 | 80.91 | 94.17 | 93.00 |
| +LoRA Tuning | 67.29 | 91.81 | 42.36 | 55.47 | 72.55 | 35.98 | 89.00 | 46.11 | 46.69 | 79.93 | 91.41 | 92.88 |
| +DDO | 72.04 | 93.25 | 27.08 | 60.66 | 80.78 | 48.32 | 88.00 | 57.84 | 53.67 | 79.94 | 97.19 | 94.74 |
| +Ours | **72.38** | 91.45 | 40.27 | 61.04 | 85.36 | 51.75 | 85.74 | 53.34 | 58.33 | 81.05 | 94.62 | 93.24 |
| Wan2.1-1.3B | 69.25 | 95.21 | 34.03 | 60.34 | 81.33 | 52.29 | 76.00 | 67.43 | 33.00 | 71.51 | 95.28 | 95.29 |
| +LoRA Tuning | 66.91 | 97.00 | 28.47 | 58.73 | 74.84 | 50.46 | 74.00 | 67.68 | 25.28 | 68.94 | 95.38 | 95.23 |
| +DDO | 69.18 | 95.42 | 32.63 | 59.94 | 77.55 | 59.22 | 75.00 | 73.50 | 25.54 | 70.86 | 95.07 | 96.25 |
| +Ours | **69.86** | 96.31 | 34.03 | 60.35 | 81.49 | 54.74 | 78.00 | 69.52 | 32.35 | 71.85 | 95.23 | 94.55 |

Table 1: Comparison of sub-dimensional scores on VBench Huang et al. (2024b) before and after optimization for CogVideoX-2B, CogVideoX-5B Yang et al. (2024), and Wan2.1-1.3B Wang et al. (2025) (CogVideoX-2B, CogVideoX-5B referred to as CogV-2B and CogV-5B, respectively).

| Model | FVD↓ | KVD↓ | crop-FVD↓ | crop-KVD↓ |
|---|---|---|---|---|
| CogV-2B Chen et al. (2024) | 449.31 | 22.75 | 232.10 | 7.77 |
| +LoRA Tuning | 552.66 | 29.27 | 253.87 | 9.53 |
| +DDO | **442.58** | 21.28 | 229.54 | 7.93 |
| +Ours | 447.25 | **20.94** | **223.98** | **7.52** |
| CogV-5B Chen et al. (2024) | 433.06 | **20.93** | 222.76 | 7.56 |
| +LoRA Tuning | 494.61 | 26.37 | 219.41 | 8.21 |
| +DDO | 455.55 | 24.33 | 234.20 | 8.18 |
| +Ours | **426.36** | 21.22 | **214.63** | **7.38** |

Table 2: Comparison of FVD, KVD, crop-FVD, and crop-KVD metrics before and after optimization, along with post-training baselines and LoRA tuning, for CogVideoX-2B and CogVideoX-5B Yang et al. (2024) (denoted as CogV-2B and CogV-5B, respectively).

model and additional experiments are provided in the supplementary material. These well-trained baselines serve as reference models in our optimization experiments. We set $K = 700$ and $w = 1.1$ for degraded data generation in all experiments, and for each real sample, we generate only one corresponding degraded version. We use LoRA Hu et al. (2021) adapter for tuning the pretrained video diffusion models for 500 steps with a global batch size of 4, using AdamW optimizer with a learning rate of $6e - 6$. The LoRA rank is 128 and LoRA alphra is 64. All experiments are conducted on 4 Nvidia A100 GPUs.

**Evaluation Metrics.** To evaluate our method and compare it with baselines, we adopt a diverse set of metrics capturing both objective quality and human perception. We use VBench Huang et al. (2024b), a widely used benchmark that provides fine-grained evaluation across 16 hierarchical dimensions of video quality and semantic alignment. In addition, we report Fréchet Video Distance (FVD) Unterthiner et al. (2018), Kernel Video Distance (KVD), and their cropped variants (crop-FVD and crop-KVD), which are computed on randomly cropped video patches, to quantify the distributional similarity between generated and real videos. To complement these metrics, we conduct a user study to assess perceptual quality and human preference. More details can be found in the supplementary material.

## 5.2 COMPARISON AND EVALUATION

**Quantitative Evaluation.** Table 2 presents the total scores of VBench results, following the same evaluation setup as VideoDPO, as well as FVD/KVD, and crop-FVD/crop-KVD. These metrics

---

[1]Here, we directly report the original results from Liu et al. (2024), as no pretrained model is publicly available and CogVideoX-5B was not included in their experiments.

capture a broad range of aspects, with particular emphasis on low-level structural fidelity, motion smoothness, and temporal consistency. For model baselines, we adopt CogVideoX-2B, CogVideoX-5B, and Wan2.1-1.3B, and evaluate our method against two most recent optimization approaches, including DDO Zheng et al. (2025), and VideoDPO Liu et al. (2024). To ensure fairness, we use the released sampled dataset for VideoDPO. For all other methods, including ours, we use the same real-world training set consisting of 2,000 high-quality video-text pairs. Our proposed optimization outperforms both the base models and all competing methods across multiple metrics, with particularly notable improvements in low-level structures. This performance gain is attributed to our self-degradation scheme, which enables the simulation of diverse degradation patterns guided by the semantics of real data. By effectively employing such fine-grained supervision from both degraded and real data, our method encourages the model to steer away from the degraded distribution and move toward higher-quality, temporally consistent generation. More comparisons are left in supplementary.

**Qualitative Evaluation.** Figure 4 presents qualitative visual comparisons between the base model (CogVideoX-5B) and our method augmented with the proposed optimization strategy. Additional examples are included in the supplementary material for a more comprehensive evaluation. As illustrated, the base model frequently suffers from motion inconsistency and structural artifacts, particularly in complex scenes involving object interactions or occlusions. For instance, in example 2 of Figure 4, the car is incorrectly rendered as splitting into two separate vehicles, indicating a failure in maintaining temporal and spatial coherence. In contrast, our method significantly reduces such artifacts, yielding more temporally stable and semantically consistent results. Furthermore, our approach improves the overall visual fidelity by producing outputs with more saturated colors and richer textures, as seen the book spines (example 3). It also better preserves low-level structures, e.g., the human's facial and hair (example 1) and the raccoon's facial features (example 4) appear sharper and more realistic than those generated by the base model. **What does our SDO improve:** Since the real-degraded sample pairs generated through our degradation simulation retain similar high-level semantic information, the post-training phase primarily focuses on correcting lower-level structural issues—particularly enhancing consistency, preserving fine details, and correcting distorted structures in dynamic video generation scenarios.

## 5.3 ABLATION STUDY

We present key ablation studies below. Additional results are in the supplementary materials due to space limitations.

**The Effect of Timestep-Aware $\beta$.** We investigate the role of the timestep-aware weighting factor $\beta$ in balancing the contribution of each timestep during the self-degradation process. Specifically, we compare our default setting with two variants: a fixed $\beta = 1$ and a randomly sampled $\beta \sim \mathcal{U}(0.5, 1.5)$. These settings yield total scores of VBench 70.76 and 70.34, respectively, compared to 71.09 achieved by our method. Using a timestep-aware $\beta$ yields more stable degradations and consistent gradients, leading to better post-training performance. This suggests that adaptively modulating the strength of degradation across different timesteps is beneficial for simulating diverse yet realistic.

**Compared to LoRA Tuning.** We compare our proposed optimization strategy with conventional LoRA tuning using standard diffusion loss. While LoRA tuning updates the same rankness and same training set, it lacks a degradation-based contrastive signal, and thus struggles to generalize to complex motion or temporal inconsistencies. As shown in Table 2, our method consistently outperforms LoRA-Tuning across all evaluation metrics, particularly in motion smoothness and temporal stability. This highlights the advantage of guiding optimization through meaningful real–degraded pairs, rather than relying solely on pixel-wise reconstruction objective.

**Compared to Random Degradation.** To further investigate the differences between our self-degradation simulation and random degradation, we also apply a random degradation setup involving frame dropping, temporal shuffling, and visual distortions such as blur and noise (specific settings are left in supplementary). This random degradation yields a VBench score of 67.34, compared to 71.09 for our method, suggesting that overly chaotic degradation makes it too easy for the model to distinguish between good and bad examples. As a result, most of the optimization still relies on high-quality videos (good examples).

## 6 CONCLUSION

In this paper, we introduced a novel approach that addresses key challenges in preference-based optimization for video generative models. By eliminating the need for human labels or external scoring models, our method provides a clear and reliable signal for improving generation quality. The proposed self-degradation scheme introduces controlled perturbations to latent representations, generating degraded samples that align with the model's distribution. This approach mitigates common issues like overfitting to score models and excessive inference times. Experimental results demonstrate that our method leads to substantial improvements in both structural and semantic quality, with minimal training data and fine-tuning, offering an efficient path for enhancing generative models, particularly in video generation.

**Limitations.** Since our alignment optimization relies on additional tuning data, its performance is highly dependent on the diversity and coverage of that data. When the tuning set lacks sufficient representation—such as in uncommon artistic styles or niche domains—the learned preference signals do not generalize well, resulting in limited improvement. For example, the model may have bad performance on stylized generations involving abstract brushwork or surreal colors, where alignment quality remains suboptimal.

## ETHICS STATEMENT

All experiments use publicly available datasets and model checkpoints. No human or animal subjects are involved. The method is intended for research only, and the authors declare no competing interests.

## REPRODUCIBILITY STATEMENT

All datasets are public, and implementation details, hyperparameters, and proofs are provided in the appendix. Code will be released to ensure full reproducibility.

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

# A APPENDIX

## A.1 MORE IMPLEMENTATION DETAILS

**Implementation Details.** We conduct a comprehensive comparison between our proposed pipeline and several state-of-the-art open-source text-to-video generation models, including CogVideoX-2B Yang et al. (2024), CogVideoX-5B Yang et al. (2024), and Wan2.1-1.3B Wang et al. (2025). For CogVideoX-2B and CogVideoX-5B, to ensure consistency across methods, we uniformly preprocess all training videos to a resolution of $480 \times 720$ with a fixed number of 49 frames per clip and a frame rate of 16 frames per second. For Wan2.1-1.3B model, we preprocess all training videos to a resolution of $480 \times 832$ with a fixed number of 89 frames per clip and a frame rate of 24 frames per second. For generating high-quality video-text training pairs, we adopt VILA Lin et al. (2023), a recently proposed video captioning model known for its accurate and diverse text descriptions.

## A.2 MORE VISUAL EXAMPLES

Please view the video example in the **attached webpage**. It is recommended to use the Chrome browser to open it. Click on the **'supp.html'** file and choose to open it with the **Chrome browser**.

## A.3 MORE ABLATION STUDIES

| $w$ | $K$ | VBench↑ |
|-----|-----|---------|
| 1.1 | 700 | 72.38 |
| 1.1 | 600 | 72.53 |
| 1.05 | 700 | 72.34 |

Table 3: Ablation study on different hyper-parameters $w$ and $K$.

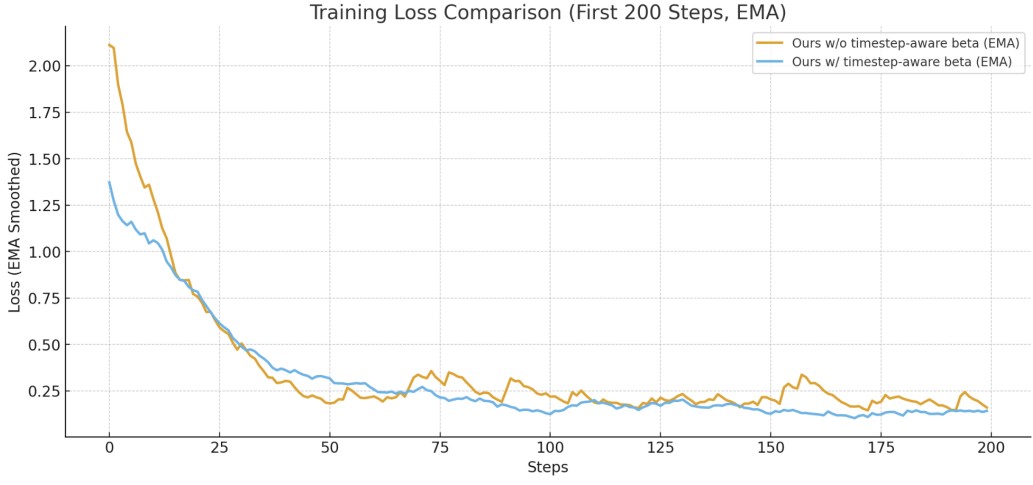

Figure 5: Comparison of training loss (EMA-smoothed) during the first 200 steps of DPO optimization. The proposed timestep-aware $\beta$ significantly stabilizes training, leading to smoother convergence and preventing gradient vanishing issues observed in the baseline.

**The effects of different $K$ and $w$.** We conduct an ablation study to investigate the effect of the weighting factor $w$ and the forwarded timestep $K$ on generation quality. As shown in Table 3 and Figure 3 in the main paper, both $w$ and $K$, when set within reasonable ranges, can enhance the results and influence the extent of degradation applied to the same high-quality video. A lower $w$ results in milder degradation, which may limit the coverage of the degradation simulation distribution in some cases. A lower $K$ leads to more similar real/degraded sample pairs, providing more accurate supervision and potentially making the post-training process more stable and easier to optimize. Due

to the current time constraints of video model sampling, it is not feasible to perform online random sampling of $w$ and $K$ to further increase the diversity of simulated degradations. We leave this for future exploration.

**The effects of time-aware beta.** As shown in Fig. 5, introducing the proposed timestep-aware $\beta$ greatly improves the stability of DPO optimization. For clearer visualization, we plot the Exponential Moving Average (EMA) of the training loss over the first 200 steps with a global batch size of 8. The baseline without timestep-aware $\beta$ suffers from strong oscillations due to unstable gradient signals across timesteps, and is prone to gradient vanishing or even collapsing during optimization. In contrast, our method dynamically stabilizes the supervision strength at different diffusion timesteps, effectively suppressing oscillatory behavior and maintaining well-conditioned gradients. As a result, the loss exhibits a much smoother convergence trajectory within fewer steps.

## A.4  USER STUDY

Video quality was evaluated through a blind user study involving over 50 participants on 50 groups of videos. Each participant was shown a pair of videos generated from the same text prompt and random seed—one by the baseline model and the other by our optimized model. The video order was randomized to prevent positional bias. Participants were asked to choose their preferred video based on various aspects, including temporal consistency, motion quality, semantic alignment with the prompt, and low-level structural fidelity. The evaluated video examples are generated by different baseline models. The evaluated video examples are generated by different baseline models. As shown in Figure 6, videos produced by our method were preferred in most cases, demonstrating consistent improvements across various dimensions of text-to-video generation—particularly in terms of temporal consistency and low-level structural fidelity.

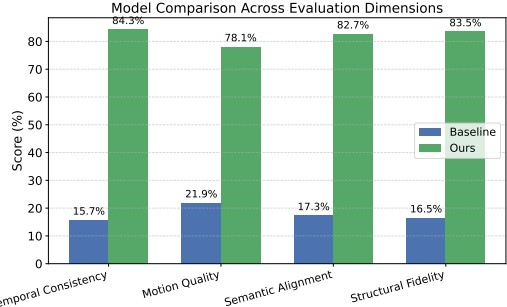

Figure 6: User study results comparing baseline models and Ours across diverse evaluation criteria.

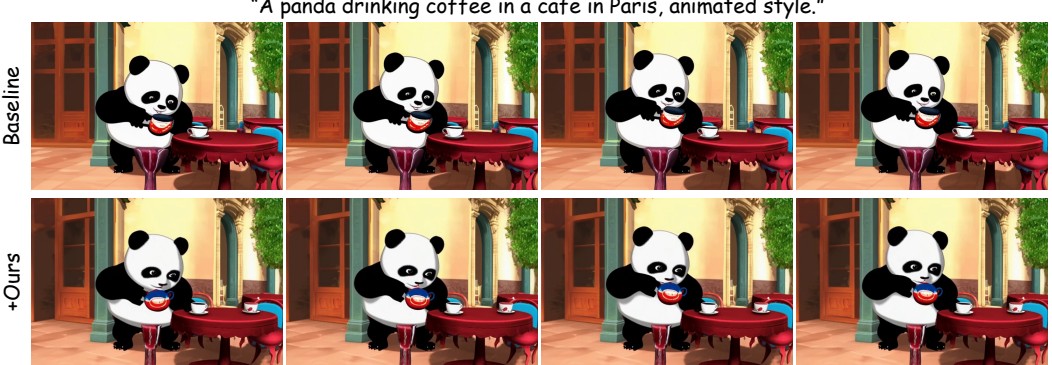

Figure 7: An example of a failure case. Performance on certain specific styles may degrade due to the lack of style diversity in our training data.

## A.5   Limitation and Failure Case

Since our alignment optimization relies on additional tuning data, its performance is highly dependent on the diversity and coverage of that data. When the tuning set lacks sufficient representation—such as in uncommon artistic styles or niche domains—the learned preference signals do not generalize well, resulting in limited improvement. For example, the model performs poorly on stylized generations involving abstract brushwork or surreal colors, where alignment quality remains suboptimal. These findings underscore the importance of incorporating more diverse examples during optimization to better capture the variability in real-world scenarios. Representative failure cases are included in the supplementary material. Figure 7 illustrates a failure case in which the generated video exhibits slight deformation in the cup shape, which is mitigated by our optimization method. The degradation in performance on certain specific styles is likely due to the limited style diversity in our training data.

## A.6   Clarification of LLM Usage

The LLM was employed exclusively for grammar correction and readability improvements. All conceptual contributions, analyses, and results were solely developed by the authors.

