# OpenReview forum: "Self-Discriminative Optimization for Video Diffusion Models"
_ICLR.cc/2026/Conference — ICLR 2026 Conference Desk Rejected Submission_

### Official Review · Reviewer_mPBn · 2025-10-30

**Soundness:** 2
**Presentation:** 1
**Contribution:** 2
**Rating:** 2
**Confidence:** 4

**Summary:**

This paper proposes Self-Discriminative Optimization (SDO), a new post-training method for video diffusion models that improves quality without human labels or reward models.
SDO introduces self-degradation, perturbing real video latents in the frequency domain to create controlled “degraded” samples that resemble low-quality generations. The model is then fine-tuned to distinguish real vs. degraded pairs, providing a stable, label-free supervision signal.
Applied to CogVideoX-2B/5B, SDO improves structural fidelity, motion consistency, and semantic alignment with minimal data (≈2k videos) and 500 LoRA steps.
Compared to LoRA tuning, DPO, and DDO, it achieves higher VBench and lower FVD/KVD scores, showing better temporal smoothness and fewer artifacts.
The method is efficient, avoids overfitting to score models, and stabilizes optimization, though its performance depends on the diversity of fine-tuning data.

**Strengths:**

- Proposes a novel label-free optimization method that learns from real videos and their degraded counterparts generated via the diffusion model itself. This design eliminates the need for human annotations, significantly reducing alignment cost while providing reliable supervision.
- Demonstrates consistent improvements across different model scales (CogVideoX-2B and 5B), showing the method’s scalability and general applicability to various diffusion backbones.

**Weaknesses:**

- The paper provides almost no quantitative results, and the evaluation relies mainly on classical metrics such as FVD, which are poorly correlated with human perceptual quality and therefore lack persuasiveness for assessing modern video generation models [1]. The only relatively new metric used is VBench, but only the total score is reported. Reporting only the total VBench score is insufficient because it is biased toward temporal consistency, meaning that results that sacrifice dynamics can still achieve deceptively high scores [2], [3], [4]. Therefore, please present all individual VBench metric values.
- The absence of any human evaluation further weakens the argument. Based on the reported quantitative results, the performance gains appear marginal from a human perception standpoint. To address this concern, the authors should conduct a Gold Human pairwise comparison experiment against prior methods, as this has become the standard in recent video generation papers [5], [6], [7] [8].
- The choice of base models is also insufficient. The paper does not discuss whether the proposed method could yield performance gains on state-of-the-art video diffusion models, such as Wan 2.1 or HunyuanVideo. Without such discussion or experiments, it remains unclear whether the method is effective and scalable to the latest, large-scale models, limiting the generality and practical significance of the results.
- Finally, the ablation studies are far from adequate. While it is clear that FFT is used to decompose frequency components and assign weights, the paper does not discuss the thresholding strategy. In addition, does this method actually achieve better performance efficiency in terms of training steps or computation cost, compared to prior methods? Moreover, it would be valuable to analyze not only the overall generation quality measured by benchmarks, but also how the proposed alignment method affects generation diversity.

[1] Ge, et al. On the Content Bias in Fr ́echet Video Distance. CVPR2024.
[2] Liao, et al. Evaluation of Text-to-Video Generation Models: A Dynamics Perspective. NeurIPS2024.
[3] Liu, et al. VideoDPO: Omni-Preference Alignment for Video Diffusion Generation. CVPR2025.
[4] Oshima, et al. Inference-Time Text-to-Video Alignment with Diffusion Latent Beam Search. NeurIPS2025.
[5] Wu, et al. Boosting Text-to-Video Generative Model with MLLMs Feedback. NeurIPS2024.
[6] Hila, et al. VideoJAM: Joint Appearance-Motion Representations for Enhanced Motion
Generation in Video Models. ICML2025.
[7] Shaulov, et al. FlowMo: Variance-Based Flow Guidance for Coherent Motion in Video Generation. NeurIPS2025.
[8] Wu, et al. DenseDPO: Fine-Grained Temporal Preference Optimization for Video Diffusion Models. NeurIPS2025.

**Questions:**

Please see the weakness.

---

> ### Author Response · Authors · 2025-11-26
> **Response to Reviewer mPBn**
>
> We sincerely appreciate your valuable feedback, which has helped us improve the manuscript. Our detailed responses are provided below.
>
> > ### **W1.1: Sacrifice dynamics?**
>
> Our post-training solution will **NOT** sacrifice dynamics. As shown in our uploaded video examples (in supplementary `supp.html`), the videos produced by our post-training method remain largely similar to the original videos; the model mainly corrects issues such as temporal inconsistencies, motion smoonthness, or incorrect structures. Therefore, the **overall motion remains essentially the same**, and the motion does not become noticeably more static.
>
> This is because the negative–positive sample pairs we construct are generally very similar (as shown in Fig. 1), providing an effective form of supervision that resembles pixel-level guidance. As a result, the post-training adjustments mainly occur in lower-level structures. In contrast, existing DPO settings often use negative and positive samples that differ substantially (comparison illustrated in Fig. 1), which tends to make training less stable.
>
> > ### **W1.2: Complete results on VBench**
>
> Below is the complete results on VBench.
>
> | Model        | Total | Motion smooth. | Dynamic degree | Aesthetic quality | Object class | Multiple objects | Human action | Spatial relation. | Scene | Appear. style | Subject consist. | Back. consist. |
> |--------------|-------|----------------|----------------|-------------------|--------------|------------------|--------------|--------------------|--------|----------------|------------------|----------------|
> | CogV-2B        | 70.48 | 92.93 | 38.89 | 60.64 | 77.92 | 46.41 | 85.31 | 52.50 | 51.53 | 80.25 | 95.16 | 93.69 |
> | +LoRA Tuning       | 64.23 | 88.87 | 38.89 | 52.80 | 69.77 | 30.86 | 82.94 | 42.17 | 36.24 | 78.60 | 94.64 | 90.80 |
> | +DDO               | 69.02 | 89.05 | 37.86 | 55.32 | 74.93 | 50.06 | 83.27 | 55.18 | 45.76 | 78.61 | 94.04 | 95.19 |
> | +VideoDPO      | 70.01 | 88.64 | 38.89 | 58.64 | 77.22 | 54.04 | 81.00 | 54.90 | 45.69 | 79.73 | 94.67 | 96.64 |
> | **+Ours**              | 71.09 | 93.62 | 38.89 | 57.95 | 79.35 | 52.13 | 85.44 | 60.31 | 47.28 | 79.46 | 95.12 | 92.42 |
> | CogV-5B        | 72.09 | 90.29 | 42.36 | 61.77 | 82.91 | 57.54 | 87.87 | 53.66 | 48.52 | 80.91 | 94.17 | 93.00 |
> | +LoRA        | 67.29 | 91.81 | 42.36 | 55.47 | 72.55 | 35.98 | 89.00 | 46.11 | 46.69 | 79.93 | 91.41 | 92.88 |
> | +DDO        | 72.04 | 93.25 | 27.08 | 60.66 | 80.78 | 48.32 | 88.00 | 57.84 | 53.67 | 79.94 | 97.19 | 94.74 |
> | **+Ours**              | 72.38 | 91.45 | 40.27 | 61.04 | 85.36 | 51.75 | 85.74 | 53.34 | 58.33 | 81.05 | 94.62 | 93.24 |
>
>
> > ### **W2: Human evaluation**
>
> Thanks for your suggestion. Following the mentioned [2-4], we conducted an additional user study to further validate the effectiveness of our results, as shown in Fig. 6 of the Appendix. The blind evaluation involved more than 50 participants and 50 sets of videos. The videos were assessed based on temporal consistency, motion quality, semantic alignment with the prompt, and low-level structural fidelity.
>
> | Evaluation Dimension      | Baseline | Ours   |
> |-------------------------|:--------:|:------:|
> | Temporal Consistency    | 15.7%    | 84.3%  |
> | Motion Quality          | 21.9%    | 78.1%  |
> | Semantic Alignment      | 17.3%    | 82.7%  |
> | Structural Fidelity     | 16.5%    | 83.5%  |
>
> > ### **W3: Experiments on more base models**
>
> Thanks for the suggestion. We have additionally conducted experiments on Wan2.1-1.3B. The updated VBench evaluation results across individual dimensions are shown below. Notably, we observe clear improvements in most metrics—particularly in motion smoothness and object-related attributes.
>
> | Model        | Total | Motion smooth. | Dynamic degree | Aesthetic quality | Object class | Multiple objects | Human action | Spatial relation. | Scene | Appear. style | Subject consist. | Back. consist. |
> |--------------|-------|----------------|----------------|-------------------|--------------|------------------|--------------|--------------------|--------|----------------|------------------|----------------|
> | Wan2.1-1.3B  | 69.25 | 95.21            | 34.03         | 60.34             | 81.33       | 52.29        | 76.00       | 67.43           | 33.00 | 71.51        | 95.28           | 95.29            |
> | +LoRA Tuning | 66.91 | 97.00 | 28.47 | 58.73 | 74.84 | 50.46 |74.00|67.68 |25.28 | 68.94 | 95.38 | 95.23|
> | +DDO | 69.18 | 95.42 | 32.63 | 59.94 | 77.55 | 59.22 | 75.00 |73.50 | 25.54  | 70.86|95.07 |96.25|
> | +Ours      | 69.86 | 96.31            | 34.03         | 60.35             | 81.49       | 54.74        | 78.00       | 69.52           | 32.35 | 71.85        | 95.23           | 94.55     |

---

> ### Author Response · Authors · 2025-11-26
> **Response to Reviewer mPBn**
>
> > ### **W4: More ablation studies**
>
> > ### **W4.1: FFT thresholding**:
>
> We follow the same setting of previous work FreeInit [1] to decompose low and high-frequency components. For the frequency perturbation weight $w$, we have included the corresponding ablation study in Table 3 of the Appendix, which show that a reasonable range of hyperparameters consistently yields improvements over the base models.
>
> **How to adjust hyper-parameters?**
> Since we designed a self-degradation process to generate positive–negative sample pairs, and this procedure is training-free and controlled only by a few hyperparameters (i.e., timestep $K$ and perturbation weight $w$), it is true that the degradation strength may vary across different scenes. However, **because real data are used as the positive reference during training, the exact severity of degradation applied to the negative samples does not require careful tuning**. We have included ablation studies on different values of $K$ and $w$ in Table 3 of the Appendix, which show that **a reasonable range of hyperparameters consistently yields improvements** over the base models.
>
> > ### **W4.2: Efficiency**.
>
> We use only 2,000 high-quality videos and train for 500 fine-tuning steps (global batch size of 4) for post-training without human annotations. A comparison of training time and data usage across competing methods is provided in the table below.
>
> | Methods     | Training data | GPU Hours (A100) |
> | ----------- | ------------- | ---------- |
> | DDO         |    2,000           |      ~16      |
> | LoRA Tuning |    2,000           |        ~16    |
> | VideoDPO    |    40,000           |      ~192      |
> | Ours        |   2,000            |     ~16       |
>
> **Why only minimal cost?**
> Our lightweight post-tuning is designed primarily to activate the capabilities already present in the base model, rather than substantially expand or improve those abilities given such a limited dataset. What we achieve is that, by relying on our constructed negative–positive sample pairs, we provide explicit failure cases that guide the model to push away from undesired behaviors during post-tuning.
>
>
>
> > ### **W4.3: Quality v.s. diversity**:
>
> As mentioned before, our overall approach does not sacrifice diversity or motion dynamics. With the same seed, the videos generated after post-training remain largely similar (general motion and scene) with the original ones. Our SDO primarily focuses on correcting structural distortions and enhancing temporal smoothness—improving these lower-level aspects without altering the high-level behaviors.
>
> Reference:
>
> [1] FreeInit: Bridging Initialization Gap in Video Diffusion Models (ECCV'24)

---

### Official Review · Reviewer_LMu8 · 2025-10-30

**Soundness:** 3
**Presentation:** 3
**Contribution:** 3
**Rating:** 4
**Confidence:** 4

**Summary:**

This paper introduces Self-Discriminative Optimization that leverages human-designed degradation schemes to as low-quality negative samples. In comparison, real data are used as positive samples. The samples are then used as the negative/positive pair for optimizing DPO objective. The entire training process does not require human label and can be more easily implemented in large scale. In addition, the paper proposes diffusion time-dependent $\beta(t)$ to better optimize DPO objective.

**Strengths:**

- The presentation is clear, and the method is clearly understandable.
- The presented method is simple and compatible with most video generative models.

**Weaknesses:**

- It is unclear what data the model is trained on. If SDO is trained on significantly better data than baseline such as VideoDPO whose score is reported from the original paper, then the gain may not be entirely due to the method itself.

- Why for CogV-5B there is no DPO baseline? Since the novelty of the work is on how one obtains the positive/negative pairs, it would be useful to ablate by swapping this with the default process DPO obtains the positive/negative pairs (human ratings).

- Scores like VBench are still not so reliable as a quality metric. How does the performance increase correlate with human ratings? For example, does performance increase in VBench mean that SDO perform better in human user study?

**Questions:**

I would like the author to address my concerns above.

---

> ### Author Response · Authors · 2025-11-26
> **Response to Reviewer LMu8**
>
> We sincerely appreciate your valuable feedback, which has helped us improve the manuscript. Our detailed responses are provided below.
>
> > ### **W1: Training data**
>
> We use approximately 2,000 high-quality videos collected by ourselves, covering diverse scenes. The corresponding dataset will be released upon acceptance. To ensure fairness and avoid data-related bias, we also apply LoRA tuning using the exact same data. However, as shown in Table 1, LoRA tuning even degrades performance on both VBench and FVD, indicating that conventional fine-tuning cannot effectively improve the model under such limited data.
>
> **Why is a small amount of data sufficient?**
>
> (1) Our lightweight post-tuning requires only ~500 iterations and a small dataset because it does not aim to learn new capabilities, but rather activates and corrects the latent abilities already embedded in the base model. By constructing negative–positive sample pairs, the model receives clear failure signals, enabling it to explicitly suppress undesired behaviors (e.g., distorted structures, temporal artifacts) during post-training.
>
> (2) The negative–positive pairs are highly similar in high-level semantics (as shown in Fig. 1), providing supervision that behaves like fine-grained, pixel-level guidance—without requiring large-scale datasets or costly human annotations. Therefore, the updates remain focused on low-level structural refinement, while preserving the model’s diversity, motion dynamics, and semantic alignment.
>
> > ### **W2: Why no DPO baseline for CogV-5B**
>
> Since the DPO results are taken from VideoDPO (CVPR'25), they only report performance for CogV-2B. One point worth noting is that larger models—those with inherently stronger capabilities—are typically more difficult to improve, where our method can still achieve gains on stronger baseline CogV-5B.
>
> We further conducted experiments on the Wan 2.1 model and observed consistent improvements across most metrics—particularly in motion smoothness and object-related attributes. For comparison, we also implemented VideoDPO under the same settings as the original paper (40,000 sampled videos and 192 A100 GPU hours).
>
> | Model        | Total | Motion smooth. | Dynamic degree | Aesthetic quality | Object class | Multiple objects | Human action | Spatial relation. | Scene | Appear. style | Subject consist. | Back. consist. |
> |--------------|-------|----------------|----------------|-------------------|--------------|------------------|--------------|--------------------|--------|----------------|------------------|----------------|
> | Wan2.1-1.3B  | 69.25 | 95.21            | 34.03         | 60.34             | 81.33       | 52.29        | 76.00       | 67.43           | 33.00 | 71.51        | 95.28           | 95.29            |
> | +LoRA Tuning | 66.91 | 97.00 | 28.47 | 58.73 | 74.84 | 50.46 |74.00|67.68 |25.28 | 68.94 | 95.38 | 95.23|
> | +DDO | 69.18 | 95.42 | 32.63 | 59.94 | 77.55 | 59.22 | 75.00 |73.50 | 25.54  | 70.86|95.07 |96.25|
> | +VideoDPO (Our implementation) | 69.32 | 96.22 | 31.25 | 60.02 | 76.50 | 58.38 | 75.00 |72.84 | 32.11  | 69.60 | 94.97 |95.67|
> | +Ours      | 69.86 | 96.31            | 34.03         | 60.35             | 81.49       | 54.74        | 78.00       | 69.52           | 32.35 | 71.85        | 95.23           | 94.55     |
>
> > ### **W3: Human evaluation**
>
> We conducted an additional user study to further validate the effectiveness of our results, as shown in Fig. 6 of the Appendix. The blind evaluation involved more than 50 participants and 50 randomly selected video groups. Participants assessed the videos along four criteria: temporal consistency, motion quality, semantic alignment with the prompt, and low-level structural fidelity.
>
> | Evaluation Dimension      | Baseline | Ours   |
> |-------------------------|:--------:|:------:|
> | Temporal Consistency    | 15.7%    | 84.3%  |
> | Motion Quality          | 21.9%    | 78.1%  |
> | Semantic Alignment      | 17.3%    | 82.7%  |
> | Structural Fidelity     | 16.5%    | 83.5%  |

---

> > ### Comment · Reviewer_LMu8 · 2025-11-27
> >
> > The authors have addressed some of my concerns. I increased my score to lean towards acceptance.

---

### Official Review · Reviewer_zRx3 · 2025-10-30

**Soundness:** 3
**Presentation:** 3
**Contribution:** 3
**Rating:** 6
**Confidence:** 4

**Summary:**

This paper proposes Self-Discriminative Optimization (SDO). It’s a label-free preference alignment framework for T2V diffusion models. Unlike DPO or DDO, which rely on human annotations or unpaired data, SDO constructs paired supervision automatically by introducing a self-degradation mechanism. The method perturbs real video samples in the high frequency domain to produce degraded counterparts while maintaining semantic correlation. The model is then fine-tuned to distinguish between real and degraded samples. Experiments on CogVideoX demonstrate improvements in VBench, FVD, and temporal consistency metrics using only a small number of high-quality samples.

**Strengths:**

1.	This paper introduces the self-degradation, which eliminates the need for human annotations or additional score models.
2.	The authors clearly identify two key challenges in prior alignment methods: overfitting to score models and limited gradient signal.
3.	Results on CogVideoX-2B/5B show notable gains over both DDO and VideoDPO, particularly in temporal consistency and low-level structure preservation.

**Weaknesses:**

1.	The effectiveness of frequency-domain reweighting has only been empirically demonstrated, lacking theoretical support or validation through human evaluation experiments. High-frequency information often contains edge and texture details; therefore, simple reweighting operations are not equivalent to “low quality.”
2.	Addressing the gradient problems is an important motivation of this paper. The authors claim that the timestep-aware β(t) improves stability, but they provide no mathematical or visual explanation, nor do they show its actual impact on the optimization process (e.g., loss curves or gradient norms). The core design of the paper, self-degradation, also appears to have no clear connection to this motivation.
3.	SDO use the fixed empirical values for hyper-parameters K and w. However, the frequency distribution of different videos varies greatly, so fixed K and w cannot adapt to such variations.
4.	The experimental setup is limited. The paper compares SDO primarily against DDO and VideoDPO. It would be better to compare to other baselines, such as RLHF-based approaches. Besides, the results of ablation study are not presented clearly.

**Questions:**

1.	Could you offer some theory analysis to validate the effectiveness of frequency-domain reweighting? Or could you provide some experimental results to prove the alignment between frequency-domain reweighting and human evaluation?
2.	What specific effects does the timestep-aware adaptive scaling factor have on the optimization process? Is there any visual evidence to support it?
3.	Why the discriminative signal between real and self-degraded pairs provides better gradients than unpaired DDO? Could you provide some theoretical discussion?
4.	If the 3DFFT is replaced with other frequency analysis methods, what impact will it have on the final performance? What’s the advantage of 3DFFT over other methods?

---

> ### Author Response · Authors · 2025-11-26
> **Response to Reviewer zRx3**
>
> We sincerely appreciate your valuable feedback, which has helped us improve the manuscript. Our detailed responses are provided below.
>
> > ### **W1: Effectiveness of frequency-domain reweighting**
>
> The frequency-domain re-weighting is applied not on $z_0$ but on the noisy latent $z_t$. Amplifying high-frequency components at this stage does NOT aim to enhance edges or textures—it introduces frequency-domain perturbations and increases randomness in the generation process (i.e., destroying the generation process). This form of self-degradation is conceptually similar to noise-rescheduling strategies used in prior training-free methods such as [1-4], which **apply low-pass filtering to suppress high-frequency components in latents and thereby reduce generative randomness/diversity** (e.g., producing smoother temporal continuity in videos or preventing uncertain textures in high-resolution image synthesis). **Our self-degradation does the opposite: instead of using noise rescheduling to improve the quality or stability of video/image generation, we deliberately use it to reduce stability and produce worse outputs**.
>
> **Benefits of our self-degradation design**:
>
> * We adopt a case-by-case, training-free strategy as a form of data augmentation to construct negative examples by reversing existing “self-enhancement’’ techniques [1-4] into self-degradation. This allows us to focus on generating worse outputs without needing to precisely control the severity of degradation.
> * Because the degradation is applied through noise rescheduling, the generated “negative’’ samples still follow the base model’s original generation distribution, enabling more faithful monitoring of model-induced artifacts.
>
> Reference:
>
> [1] FreeInit: Bridging Initialization Gap in Video Diffusion Models (ECCV'24)
>
> [2] FreeU: Free Lunch in Diffusion U-Net (CVPR'24)
>
> [3] FouriScale: A Frequency Perspective on Training-Free High-Resolution Image Synthesis (ECCV'24)
>
> [4] FreeScale: Unleashing the Resolution of Diffusion Models via Tuning-Free Scale Fusion (ICCV'25)
>
> > ### **W2: Timestep-aware β(t)**
>
> Thank you for pointing out the unclear part. We have included the loss curve comparisons in Fig. 5 of Appendix to support our time-aware β design. In practice, standard DPO training is highly susceptible to gradient vanishing and training collapse. Our two proposed techniques—the self-degradation strategy for constructing positive–negative pairs and the time-aware β weighting—are designed to improve DPO training stability, provide more accurate supervision signals, and enable faster convergence.
>
> > ### **W3: Fixed K and w?**
>
> Since we designed a self-degradation process to generate positive–negative sample pairs, and this procedure is training-free and controlled only by a few hyperparameters (i.e., timestep $K$ and perturbation weight $w$), it is true that the degradation strength may vary across different scenes. However, **because real data are used as the positive reference during training, the exact severity of degradation applied to the negative samples does not require careful tuning**. We have included ablation studies on different values of $K$ and $w$ in Table 3 of the Appendix, which show that **a reasonable range of hyperparameters consistently yields improvements** over the base models.
>
> > ### **W4: Compared to RLHF methods**
>
> We have compared our method with VideoDPO (CVPR'25), which is representative of reward-model–based approaches, as it employs a score model to label positive and negative examples based on vision–language alignment and quality metrics from existing benchmarks for preference optimization. We include a direct comparison with VideoDPO under the same baseline and training settings to ensure fairness. The results show that our approach achieves stronger performance while using the same backbone and optimization configuration. Other existing methods [5-6] unify one-step distillation with post-training, making their settings fundamentally different from ours and therefore not directly comparable.
>
> Reference:
>
> [5] T2V-Turbo-v2: Enhancing Video Generation Model Post-Training Through Data, Reward, and Conditional Guidance Design (ICLR'25)
>
> [6] Diffusion Adversarial Post-Training for One-Step Video Generation (ICML'25)

---

### Official Review · Reviewer_qMKW · 2025-11-08

**Soundness:** 3
**Presentation:** 2
**Contribution:** 3
**Rating:** 6
**Confidence:** 2

**Summary:**

This paper proposes Self-Discriminative Optimization (SDO) to fine-tune video diffusion models. SDO first yielding degrades real samples by frequency-domain reweighting, and then uses these real/degraded pairs as
positive and negative examples of DPO to optimize vido diffusion models. SDO demonstrate substantial gains
in structural quality and semantic alignment than DPO and LoRA fine-tuning.

**Strengths:**

1. This present clear motivation and the corresponding method  to deal  with the problem  of fine-tuning video diffusion models.

2.  Self-degradation is a simple but effective method to generate positive and negative paris.

3. Experimental results demonstrate SDO achieves superior performance than the existing
post-training methods, with only a handful of high-quality
samples and minimal fine-tuning

**Weaknesses:**

1. The generation of positive and negative pairs is core step of DPO. Although self-degradation is a effective method to yield these pairs, other approaches using reward model, e.g.,MLLM, ImageReward,HPSv2,  to classify the generative images as positive and negative samples   can also achieve this target. This process maybe cost more time, but many acceleration methods are able to quickly generate samples. I would like to see such comparison in the paper.

2. The authors claim they use minimal fine-tuning cost, but the comparison with respect to  training hours and traning data is missing.

3. How does SDO determine the papameters of self-degradation? What  if high-frequency components are reserved?

**Questions:**

V-bench is unable to evaluate the motion extent of generated videos. The  static videos usually get high score. How do authors measure the quality of video motion?

---

> ### Author Response · Authors · 2025-11-26
> **Response to Reviewer qMKW**
>
> We sincerely appreciate your valuable feedback, which has helped us improve the manuscript. Our detailed responses are provided below.
>
> > ### **W1: Compared to reward model**
>
> We have compared our method with VideoDPO (CVPR'25), which is representative of reward-model–based approaches, as it employs a score model to label positive and negative examples based on vision–language alignment and quality metrics from existing benchmarks for preference optimization. We include a direct comparison with VideoDPO under the same baseline and training settings to ensure fairness. The results show that our approach achieves stronger performance while using the same backbone and optimization configuration. Other existing methods [1–2] unify one-step distillation with post-training, making their settings fundamentally different from ours and therefore not directly comparable.
>
> Reference:
>
> [1] T2V-Turbo-v2: Enhancing Video Generation Model Post-Training Through Data, Reward, and Conditional Guidance Design (ICLR'25)
>
> [2] Diffusion Adversarial Post-Training for One-Step Video Generation (ICML'25)
>
> > ### **W2: Comparison on training time and data**
>
> We use only 2,000 high-quality videos and train for 500 fine-tuning steps (global batch size of 4) for post-training without human annotations. A comparison of training time and data usage across competing methods is provided in the table below. **Our lightweight post-tuning is designed primarily to activate the capabilities already present in the base model, rather than substantially expand or improve those abilities given such a limited dataset.**
>
> | Methods     | Training data | GPU Hours (A100) |
> | ----------- | ------------- | ---------- |
> | DDO         |    2,000           |      ~16      |
> | LoRA Tuning |    2,000           |        ~16    |
> | VideoDPO    |    40,000           |      ~192      |
> | Ours        |   2,000            |     ~16       |
>
>
> > ### **W3: Hyper-parameters of self-degradation**
>
> Since we designed a self-degradation process to generate positive–negative sample pairs, and this procedure is training-free and controlled only by a few hyperparameters (i.e., timestep $K$ and perturbation weight $w$), it is true that the degradation strength may vary across different scenes. However, **because real data are used as the positive reference during training, the exact severity of degradation applied to the negative samples does not require careful tuning**. We have included ablation studies on different values of $K$ and $w$ in Table 3 of the Appendix, which show that **a reasonable range of hyperparameters consistently yields improvements** over the base models.
>
> > ### **Q1: Measure of video motion**
>
> We have provided FVD measurements and a user study to further support the evaluation. In addition, as shown in our uploaded video examples (in supplementary `supp.html`), the videos produced by our post-training method remain largely similar to the original videos; the model mainly corrects issues such as temporal inconsistencies, motion smoonthness, or incorrect structures. Therefore, the overall motion remains essentially the same, and **the motion does **NOT** become noticeably more static.**
> This is also one of the advantages of designing such negative–positive sample pairs: it provides **an effective form of supervision that is similar to 'pixel-level' guidance.** (compared to existing DPO settings, where the negative and positive samples may differ substantially, as shown in Fig. 1, leading to more unstable training.)

---

### Comment · Area_Chair_WnPk · 2025-11-27
**Please check the rebuttal**

Dear Reviewers,

The authors have posted their rebuttal. Could you please take a moment to review their responses and check whether your concerns have been adequately addressed if you have done it yet? If possible, kindly initiate the discussion at your earliest convenience.

Your timely assistance is essential for keeping the review process on track. Thank you very much for your support and contribution.

Best regards,
Your AC

---

### Note · Program_Chairs · 2026-01-17
**Submission Desk Rejected by Program Chairs**

The following references in this submission do not refer to real documents and/or have major errors in bibliographic information:

 Meng Xu, Xihui Li, Yixiao Zhou, and Fisher Yu. Using human feedback to fine-tune diffusion models without any labels. arXiv preprint arXiv:2305.15013, 2023.